# Exploring Neural Architecture Search For Language Tasks

## Abstract

Neural architecture search (NAS), the task of finding neural architectures automatically, has recently emerged as a promising approach for discovering better models than ones designed by humans alone. However, most success stories are for vision tasks and have been quite limited for text, except for a small language modeling datasets. In this paper, we explore NAS for text sequences at scale, by first focusing on the task of language translation and later extending to reading comprehension. We conduct extensive searches over the *recurrent cells* and *attention similarity functions* for standard sequence-to-sequence models across two translation tasks, IWSLT English-Vietnamese and WMT German-English. We report challenges in performing cell searches as well as demonstrate initial success on attention searches with translation improvements over strong baselines. In addition, we show that results on attention searches are transferable to reading comprehension on the SQuAD dataset.

## 1 Introduction

There has been vast literature on finding neural architectures automatically dated back to the 1980s with genetic algorithms (Schaffer et al., 1992) to recent approaches that use random weights (Saxe et al., 2011), Bayesian optimization (Snoek et al., 2012), reinforcement learning (Zoph & Le, 2017; Baker et al., 2017; Zhong et al., 2017), evolution (Real et al., 2017), and hyper networks (Brock et al., 2017). Among these, the approach of neural architecture search (NAS) using reinforcement learning by Zoph & Le (2017), barring computational cost, has been most promising, yielding state-of-the-art performances on several popular vision benchmarks such as CIFAR-10 and ImageNet (Zoph et al., 2017). Building on NAS, others have found better optimizers (Bello et al., 2017) and activation functions (Ramachandran et al., 2017) than human-designed ones. Despite these success stories, most of the work mainly focuses on vision tasks, with little attention to language ones, except for a small language modeling task on the Penn Tree Bank dataset (PTB) in (Zoph & Le, 2017).

This work aims to bridge that gap by exploring neural architecture search for language tasks. We start by applying the approach of (Zoph & Le, 2017) to neural machine translation (NMT) with sequence-to-sequence (Sutskever et al., 2014) as an underlying model. Our goal is to find new recurrent cells that can work better than Long Short-term Memory (LSTM) (Hochreiter & Schmidhuber, 1997). We then introduce a novel "stack" search space as an alternative to the fixed-structure tree search space defined in (Zoph & Le, 2017). We use this new search space to find similarity functions for the attention mechanism in NMT (Bahdanau et al., 2015; Luong et al., 2015b). Through our extensive searches across two translation benchmarks, small IWSLT English-Vietnamse and large WMT German-English, we report challenges in performing cell searches for NMT and demonstrate initial success on attention searches with translation improvements over strong baselines.

Lastly, we show that the attention similarity functions found for NMT are transferable to the reading comprehension task on the Stanford Question Answering Dataset (SQuAD) (Rajpurkar et al., 2016b), yielding non-trivial improvements over the standard dot-product function. Directly running NAS attention search on SQuAD boosts the performance even further.

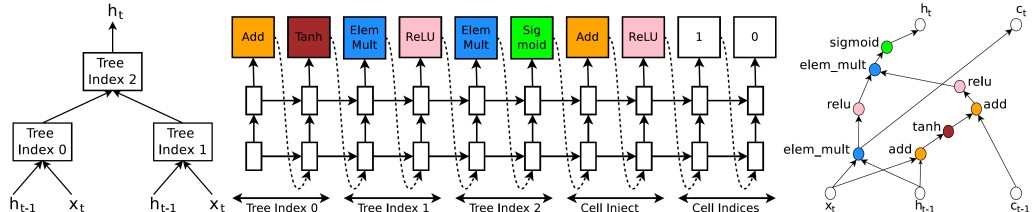

Figure 1: **Tree search space for recurrent cells** – shown is an illustration of a tree search space specifically designed for searching over LSTM-inspired cells. The figure was obtained from (Zoph & Le, 2017) with permission. *Left*: the tree that defines the computation steps to be predicted by controller. *Center*: an example set of predictions made by the controller for each computation step in the tree. *Right*: the computation graph of the recurrent cell constructed from example predictions of the controller.

## 2 NEURAL ARCHITECTURE SEARCH FOR LANGUAGES

In neural architecture search (NAS), a controller iteratively samples a model architecture which is then run against a task of interest to obtain a reward signal. The reward signal is then used to update the controller so as to produce better and better architectures over time. We follow the architecture search setup in (Zoph & Le, 2017), which was originally developed for a small language modeling task on PTB, and adapt it to other language tasks. We first focus on translation and conduct searches on both small and large scale translation tasks (IWSLT and WMT respectively, with details in Section 3). In terms of reward functions, we have a choice of using either perplexity or BLEU scores which have been known to be well-correlated in neural machine translation (Luong et al., 2015b). Formally, we scale the reward scores to be within $[0, 1]$ as follows[1]:

$$R(\text{score}) = \begin{cases} \left(\frac{\alpha}{\text{score}}\right)^\beta & [\text{score} = \text{perplexity}] \\ \left(\frac{\text{score}}{\alpha}\right)^\beta & [\text{score} = \text{BLEU}] \end{cases} \qquad (1)$$

We now describe the search spaces over recurrent cells and attention similarity functions.

### 2.1 TREE-BASED SEARCH SPACE

This search space is identical to what was designed by Zoph & Le (2017) to search for LSTM-inspired cells. Each instance of the search space takes as inputs (state $h_{t-1}$, cell $c_{t-1}$, input $x_t$) and produces as outputs (state $h_t$, cell $c_t$). The idea is to define computation through a balanced binary tree which we start off first by connecting each leaf node with inputs $(h_{t-1}, x_t)$ and producing output $h_t$ at the top node. The left part of Figure 1 illustrates a binary computation tree of depth 2. The RNN controller will then decide for each node in the tree what *combined* operations, e.g., add or mul, to use and what *nonlinearity*, e.g., tanh or sigmoid, to be immediately followed. The controller will also decide how to incorporate ("cell inject") the previous cell $c_{t-1}$ and which nodes ("cell indices") in the tree to use for that injection as well as the output of the new cell. The RNN controller is illustrated in the center part of Figure 1 and the realization of an LSTM-inspired instance is shown on the right part. The parameters of instance is defined by linear transformation of each inputs $(h_{t-1}, x_t)$ before passing to each leaf node.

**LSTM-inspired Cells for NMT**   In this work, we are interested in how this search space, which works well for the language modeling task, performs on translation. Our set of combined operations are element-wise addition and multiplication; whereas the set of non-linear functions are $(\text{identity}, \tanh, \text{sigmoid}, \text{relu})$. We use a binary tree of depth 4 with 8 leaf nodes.

---

[1]We fix $\beta = 2$ in our experiments. For $\alpha$, if the reward type is perplexity, we set to 10 for IWSLT and 48 for WMT. If the reward type is BLEU, we set to 30 for IWSLT and 15 for WMT.

## 2.2 STACK-BASED SEARCH SPACE

We propose a simple stack-based programming language as an alternative to the fixed structure of the previous tree search space. This is reminiscent of the Push language which is historically used in genetic programming (Spector et al., 2005). A program consists of a sequence of function calls, with each function popping $N$ arguments from the top of the stack and pushing $M$ results back onto it. As illustrated in Figure 2, we consider in this work only unary or binary ops, i.e., $N = 1$ or 2, with $M = 1$. If there are not enough arguments in the stack, such as a binary operation when there is a single item in the stack, the operation is ignored. This ensures every program is valid without additional constraints on the controller.

The program is also given the ability to copy outputs produced within $S$ steps ago. Such capability is achieved through ops $(\text{copy}_0, \ldots, \text{copy}_{S-1})$ which the controller can predict together with other unary and binary ops. The indices in Figure 2 indicate the order in which outputs were generated so copy ops can be applied. At the start of execution, input arguments of the same shape and data type are both pushed onto the stack. At the end of execution, we either sum all remaining values or take the top value of the stack as the output.

$$0.\ \mathbf{x} \xrightarrow{\text{copy}_0} \begin{array}{c}1.\ \mathbf{x}\\0.\ \mathbf{x}\end{array} \xrightarrow{\text{linear}} \begin{array}{c}2.\ \mathbf{Wx}\\0.\ \mathbf{x}\end{array} \xrightarrow{\text{sigmoid}} \begin{array}{c}3.\ \boldsymbol{\sigma}\mathbf{(Wx)}\\0.\ \mathbf{x}\end{array} \xrightarrow{\text{add}} 4.\ \mathbf{x + \boldsymbol{\sigma}(Wx)}$$

Figure 2: **General stack search space** – shown is an illustration of how the stack changes over a sequence of operations. In this example, the controller predicts $(\text{copy}_0, \text{linear}, \text{sigmoid}, \text{add})$ as a sequence of operations to be applied. The stack has a vector $\boldsymbol{x}$ as an initial input and produces $\boldsymbol{x} + \sigma(\boldsymbol{Wx})$ as an output.

**Attention Search for NMT**   As attention mechanism is key to the success of many sequence-to-sequence models, including neural machine translation (Bahdanau et al., 2015), it is worthwhile to improve it. To recap, we show in Figure 3 an instance of the attention mechanism proposed by Luong et al. (2015a) for NMT. At each time step, the current target hidden state $\boldsymbol{h}_t$ (a "query") is used to compare with each of the source hidden states $\bar{\boldsymbol{h}}_s$ (memory "keys") to compute attention weights which indicate which memory slots are most useful for the translation at that time step. The attention weight for each source position $s$ is often computed as $\frac{\exp\left(\text{score}(\boldsymbol{h}_t, \bar{\boldsymbol{h}}_s)\right)}{\sum_{s'=1}^{S} \exp\left(\text{score}(\boldsymbol{h}_t, \bar{\boldsymbol{h}}_{s'})\right)}$, with $S$ being the source length. In (Luong et al., 2015a), various forms of the scoring function have been proposed, e.g., the simple dot-product $\bar{\boldsymbol{h}}_s^\top \boldsymbol{h}_t$ or the bilinear form $\bar{\boldsymbol{h}}_s^\top \boldsymbol{W} \boldsymbol{h}_t$.

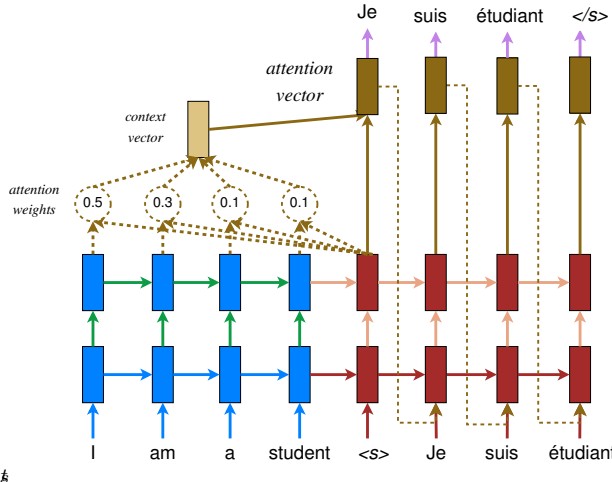

Figure 3: **Attention Mechanism** – example of an attention-based NMT system as described in (Luong et al., 2015a). We highlight in detail the first step of the attention computation.

Instead of hand-designing these functions, we propose to search through the set of scoring functions $\text{score}(\boldsymbol{q}, \boldsymbol{k})$ using our stack-based search space. The stack starts out with a key vector $\boldsymbol{k}$ followed by a query vectory $\boldsymbol{q}$ on top. The RNN controller predicts a list of $L$ ops, followed by a special reduce op that is used to turn vectors into scalars by summing over the final dimension as illustrated in Figure 4. $L$ is the program length, which controls the search space complexity and is set to 8 in our experiments. The set of binary and unary ops include $(\text{linear}, \text{mul}, \text{add}, \text{sigmoid}, \text{tanh}, \text{relu}, \text{identity})$ with mul being element-wise multiplication. The occurrence of op identity is for the controller to shorten the program as needed.

Figure 4: **Stack Search Space for Attention** – shown is the bilinear scoring function (Luong et al., 2015a) as an instance of the stack search space for attention. The controller predicts $(\text{linear}, \text{mul})$ as ops. All scoring functions end with a reduce op that turns vectors into scalars.

## 2.3 ARCHITECTURE SEARCH FOR READING COMPREHENSION

Attention mechanisms are a core part of modern question answering systems. In the extractive problem setting, given a query sentence and a context paragraph, the task is to output start and end positions in the text, called a span, which contains the answer. On the most common dataset of this type, Stanford Question Answering Dataset (SQuAD) (Rajpurkar et al., 2016a), the top performing models all perform some variant of attention between the encoded query sentence and encoded context paragraph. Many of the best models are variants of a model known as Bidirectional Attentive Flow (BiDAF) (Seo et al., 2016). BiDAF encodes the query and context separately, then performs bidirectional attention from the query vectors over the context vectors and from the context vectors over the query vectors. The similarity is generally computed as the dot product between the two vectors. To test the generalization of our attention functions and search, we apply the best attention mechanisms from NMT in place of the dot product. We also repeat the search procedure directly on the SQuAD dataset. The search is similar as for NMT, with the addition of a unary separable 1D convolution operator.

## 3 EXPERIMENTS

We consider two translation setups, *small* and *large*-scale ones, to test different effects of Neural Architecture Search (NAS). Remember that, in NAS, there are often two phases involved: (a) *search architectures* – in which we train many *child* models to find best architectures according to the reward signals and (b) *run convergence* — where we take top architectures found and train *full* models until convergence. To make things clear, for each of the translation setup below, we will describe the data used, as well as the training hyperparameters for both the child and the full models. Our evaluation metrics include both perplexity and BLEU (Papineni et al., 2002) scores; whereas, the NAS reward signals can be either perplexity or BLEU, which we will detail later.

### 3.1 SMALL-SCALE TRANSLATION (IWSLT)

**Data** We utilize a small parallel corpus of English-Vietnamese TED talks (133K sentence pairs), provided by the IWSLT Evaluation Campaign (Cettolo et al., 2015). Following Luong & Manning (2015), we tokenize with the default Moses tokenizer and replace words whose frequencies are less than 5 by `<unk>`.[2] The final data has vocabulary sizes of 17K for English and 7.7K for Vietnamese. We use the TED tst2012 (1553 sentences) as a validation set for hyperparameter tuning and TED tst2013 (1268 sentences) as a test set.

**Full-model hyperparameters** Each full model is an attention-based sequence-to-sequence model with 2 LSTM layers of 512 units each; the encoder is bidirectional and the embedding dimension is

---

[2]The processed data was downloaded from `http://nlp.stanford.edu/projects/nmt`.

also 512. We train each child model for 12K steps (roughly 12 epochs) with dropout rate of 0.2 (i.e., keep probability 0.8) and batch size 128. Our optimizer is SGD with learning rate 1; after 8K steps, we start halving learning rate every 1K step. We use 5 for gradient norm clipping and uniformly initialize the parameters within $[-0.1, 0.1]$. The exact implementation and setup were obtained from the NMT tutorial (Luong et al., 2017).

**Child-model hyperparameters**   Since this dataset is small, the hyperparameters of the child models for NAS are identical to those of the full models.

## 3.2   LARGE-SCALE TRANSLATION (WMT)

**Data**   We consider the WMT German-English translation task with 4.5M training sentence pairs. The data is split into subword units using the BPE scheme (Sennrich et al., 2016) with 32K operations. We use newstest2013 (3000 sentences) as a development set and report translation performances on both newstest2014 (2737 sentences) and newstest2015 (2169 sentences).[3]

**Full-model hyperparameters**   We train strong translation models based on the architecture of Google's Neural Machine Translation systems (Wu et al., 2016) with a implementation in (Luong et al., 2017). The model consists of 4 LSTM layers of 1024 units (the encoder starts with a bidirectional LSTM followed by three unidirectional layers); embedding dim is 1024. The hyperparameters are similar to those of the English-Vietnamese setup: init range $[-0.1, 0.1]$, dropout 0.2, gradient norm 5.0. We train with SGD for 340K steps (10 epochs). The learning rate is 1.0; after 170K steps, we start halving learning rate every 17K step.

**Child-model hyperparameters**   Since we cannot afford to run NAS on full models, our child model is a scaled-down version of the full one with 2 layers and 256 units, trained for 10K steps.

## 3.3   TRAINING OF THE CONTROLLER

Following Zoph et al. (2017), we train the controller RNN using the Proximal Policy Optimization (Schulman et al., 2015) with a learning rate of 0.0005. To encourage exploration, an entropy penalty 0.0001 was used. We use an exponential moving average of previous rewards with weight 0.99 as a baseline function. The controller weights are uniformly initialized within $[-0.1, 0.1]$. We use minibatches of 20 architectures to update the controller RNN weights.

During search, we employ a global workqueue system similar to (Zoph et al., 2017) to process a pool of child networks proposed by the RNN controller. In our experiments, the pool of workers consists of 100-200 GPUs. We stop the searches once the dev performance saturates and the set of unique child models remains unchanged. Once the search is over, the top 10 architectures are then chosen to train until convergence.

## 3.4   MAIN RESULTS

We present in Table 1 results of neural architecture search for translation. We compare over strong baselines provided by Luong et al. (2017), which replicate Goole's NMT architectures (Wu et al., 2016). As we can see in the first three rows, the strong baseline trained with SGD and LSTM as the basic unit outperforms NASCell, the TensorFlow public implementation of the best recurrent cell found on the PTB language modeling task by Zoph & Le (2017).[4] Architecture searches directly on translation tasks yield better performances compared to NASCell. We can find cells that outperform the baseline in the IWSLT benchmark with 26.2 BLEU. Beating the strong baseline on the larger WMT task remains a challenge; with cell searches performed directly on WMT, we can narrow the gap with the baseline, achieving 28.4 BLEU. Finally, by performing attention searches on WMT, we were able to outperform the WMT baseline with 29.1 BLEU. The same attention function found is also transferable to the small IWSLT benchmark, yielding a high score of 26.0 BLEU.

---

[3]The data processing script can be found at `https://github.com/tensorflow/nmt#wmt-german-english`.

[4]`https://www.tensorflow.org/api_docs/python/tf/contrib/rnn/NASCell`

| NMT Systems | IWSLT (small) | WMT (large) |
|---|---|---|
| | tst2013 | newstest2015 |
| Google NMT baseline [adam] (Luong et al., 2017) | 21.8 | 26.8 |
| Google NMT baseline [sgd] (Luong et al., 2017) | **25.5** | **28.8** |
| NASCell on LM [sgd] (Zoph & Le, 2017) | 25.1 | 27.7 |
| NASCell on IWSLT [ppl, sgd] (this work) | **26.2** | 27.9 |
| NASCell on WMT, [ppl, adam] (this work) | 23.4 | $<11$ |
| NASCell on WMT, [ppl, sgd] (this work) | 25.4 | **28.4** |
| NASAttention on WMT [ppl, sgd] (this work) | 25.9 | $<20$ |
| NASAttention on WMT [bleu, sgd] (this work) | **26.0** | **29.1** |

Table 1: **Neural architectural searches for translation** – shown are translation performances in BLEU for various neural architecture searches (NAS) at the cell and attention levels. Searches are performed on either the *small* IWSLT or *large* WMT translation setups with reward functions being either *ppl* (perplexity) or *BLEU*. For each NAS search, we report results on both translation setups. For NASCell, we use the TensorFlow public implementation by Zoph & Le (2017) and run on our translation setups. We highlight in bold numbers that are best in each group.

## 4 ANALYSIS

In this section, we continue results in Table 1 to further discuss the effects of optimizers and reward functions used in architecture search. We also show the top attention functions found by NAS and their effects. Lastly, we examine the transferability of these attention functions and searches to the task of reading comprehension.

### 4.1 EFFECTS OF OPTIMIZERS FOR NMT MODELS

We found that optimizers used for NMT models matter greatly in architecture search. From the training plots in Figure 5, it seems to appear that Adam outperforms SGD greatly, achieving much higher BLEU scores on both the dev and test sets after a fixed training duration of 10K steps per child model. However, as observed in rows 5 and 6 of Table 1, recurrent cells found by Adam are unstable, yielding much worse performance compared to those found by SGD. We have tried using Glorot initialization scheme (Glorot & Bengio, 2010) but could not alleviate the problem of large gradients when using Adam-found cells. We suspect further hyperparameter tuning for final-model training might help.

### 4.2 EFFECTS OF REWARD FUNCTIONS FOR ATTENTION SEARCHES

We also carry a small experiment comparing the reward functions described in Eq (1) for the attention search. From Figure 6, the reward function based on BLEU trivially leads to higher dev and test BLEU scores during the search. The attention functions found does transfer to higher BLEU scores as shown in row 8 of Table 1. What surprised us was the fact that the attention mechanisms found with perplexity-based reward function perform poorly.

The top-performing attention similarity functions found are: $\mathrm{reduce}(\mathrm{sigmoid}(\mathrm{relu}(\tanh(\boldsymbol{W}(\boldsymbol{k}\odot\boldsymbol{q})))))$ and $\mathrm{reduce}(\mathrm{sigmoid}(\mathrm{relu}(\boldsymbol{W}(\boldsymbol{k}\odot\boldsymbol{q}))))$. At first, the equations look puzzling with multiple nonlinearity functions stacked together, which we think due to noise in the design of the search space that lets the controller favor over nonlinear functions. However, a closer look does reveal an interesting pattern that keys and queries are encouraged to interact element-wise, followed by linear transformation, nonlinearity, before the final reduce-sum op. On the other hand, several bad attention functions have the following pattern of applying nonlinearity immediately after element-wise multiplication, e.g., $\mathrm{reduce}(\mathrm{sigmoid}(\boldsymbol{W}(\tanh(\tanh(\boldsymbol{k}\odot\boldsymbol{q})))))$. Nevertheless, we think the search space could have been improved by predicting when a program can end and when to perform reduce operations.

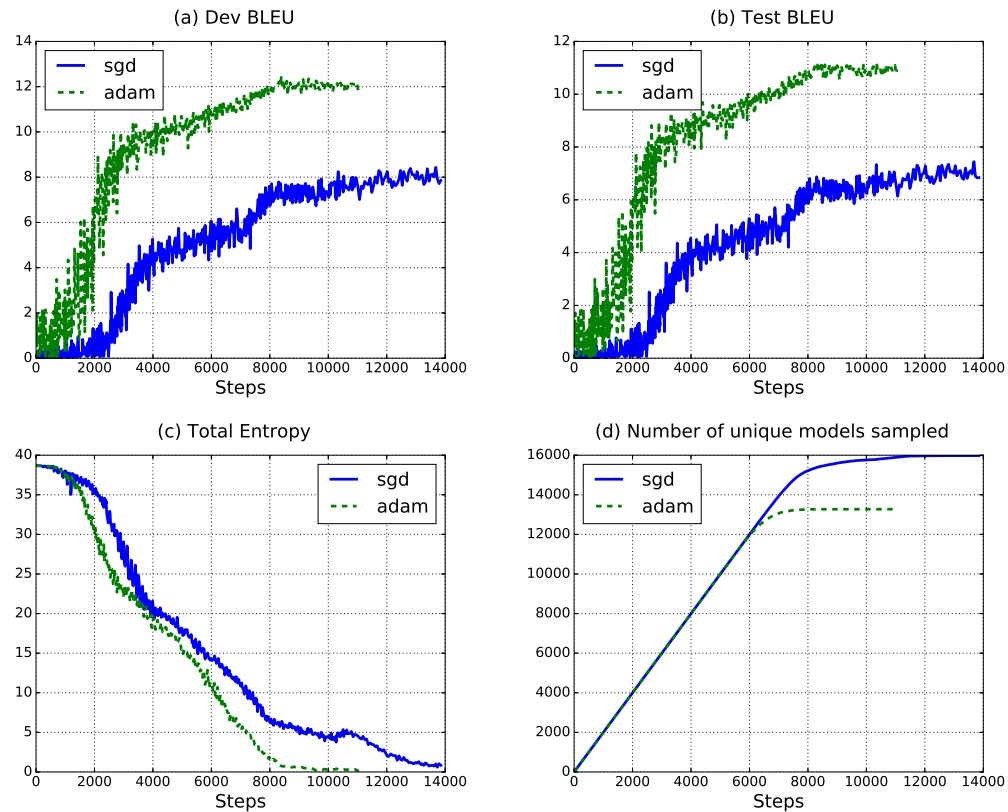

Figure 5: **Effects of optimizers for NMT models** – shown are training plots comparing two recurrent cell searches, which are different in terms of optimizers used for NMT models, *adam* or *sgd*. The four different subplots are (a, b) BLEU scores on the dev and test sets, (c) the total entropy of the RL controller, and (d) how many unique child models we have sampled for each search.

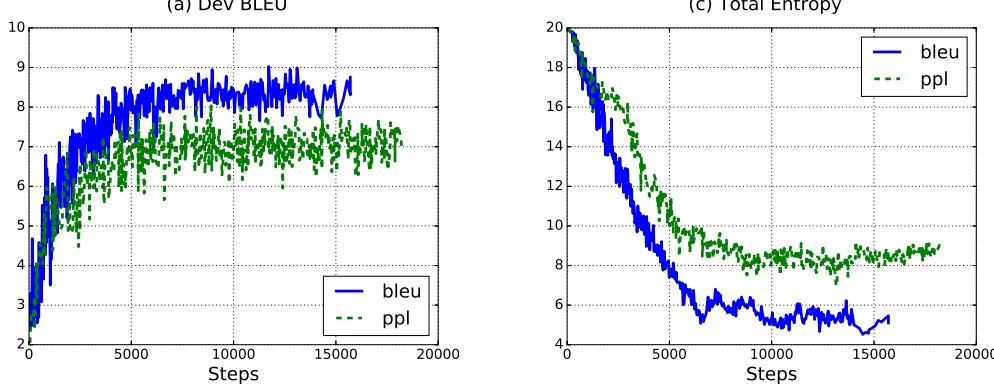

Figure 6: **Effects of reward functions for attention searches** – shown are similar plots to those in Figure 5 with two attention searches that differ in terms of reward functions used: one based one BLEU while the other is based on perplexity (pp). For brevity, we only show plot for dev BLEU and total entropy.

| SQuAD Systems | $F_1$ |
|---|---|
| BiDAF (Seo et al., 2017) | 77.3 |
| Our baseline (dot-product) | 80.1 |
| NASAttention on NMT | 80.5 |
| NASAttention on SQuAD | **81.1** |

Table 2: **Attention functions for SQuAD systems**

### 4.3 Transferability to Reading Comprehension

For the reading comprehension setting, we evaluate on the Stanford Question Answering dataset as discussed in 2.3. We report results in $F_1$ which measures the portion of overlap tokens between the predicted answer and groundtruth.

**Model** Our model details are as follows: we embed both the query and context with pretrained GLoVE (Pennington et al., 2014) embeddings. Each is then encoded independently using the *embedding encoder*, sharing weights between the two. We then combine them with context-to-query attention (leaving out the query-to-context attention from BiDAF), with the output from each position in the context the result of attending over the query with that position's encoding. When doing attention search, we search over the form of the equation to compute this similarity.

Finally, we take the output of this attention mechanism and run through a stack of three *model encoder*, giving three outputs $x_1$, $x_2$, and $x_3$, each of which is the length of the context sentence. The probability of the span starting at each position is computed as $W_0[x_1, x_2]$, and the probability of it being the end position is $W_1[x_1, x_3]$. We score each span as the product of its start position probability and end position probability, returning the span with the highest score.[5]

**Results** Table 2 demonstrates that the NASAttention found with NMT does provide improvement over the baseline with dot-product attention, yielding a $F_1$ score of $80.5$. When performing attention search directly on SQuAD, the performance is further boosted to $80.1$ $F_1$. We find that the best performing attention function for the context-to-query attention is simple yet novel: $f(key, query) = relu(conv(key \circ query))$, where $conv$ is a 1d separable convolution with a kernel size of 3. Neighboring positions in the convolution correspond to neighboring keys (in this case in the encoded question) being queried with the same query vector.

## 5 Conclusion

In this paper, we have made a contribution towards extending the success of neural architecture search (NAS) from vision to another domain, languages. Specifically, we are first to apply NAS to the tasks of machine translation and reading comprehension at scale. Our newly-found recurrent cells perform better on translation than previously-discovered NASCell (Zoph & Le, 2017). Furthermore, we propose a novel stack-based search space as a more flexible alternative to the fixed-structure tree search space used for recurrent cell search. With this search space, we find new attention functions that outperform strong translation baselines. In addition, we demonstrate that the attention search results are transferable to the SQuAD reading comprehension task, yielding non-trivial improvements over dot-product attention. Directly running NAS attention search on SQuAD boosts the performance even further. We hope that our extensive experiments will pave way for future research in NAS for languages.

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
