# OpenReview forum: "EXPLORING NEURAL ARCHITECTURE SEARCH FOR LANGUAGE TASKS"
_ICLR.cc/2018/Conference — Reject_

### Official Review · AnonReviewer2 · 2017-11-23

**Rating:** 3
**Confidence:** 4

**Review:**

The paper explores neural architecture search for translation and reading comprehension tasks. It is fairly clearly written and required a lot of large-scale experimentation. However, the paper introduces few new ideas and seems very much like applying an existing framework to new problems. It is probably better suited for presentation in a workshop rather than as a conference paper.

A new idea in the paper is the stack-based search. However, there is no direct comparison to the tree-based search. A clear like for like comparison would be interesting.

Methodology. The test set newstest2014 of WMT German-English officially contains 3000 sentences. Please check http://statmt.org/wmt14.
Also, how stable are the results you obtain, did you rerun the selected architectures with multiple seeds? The difference between the WMT baseline of 28.8 and your best configuration of 29.1 BLEU can often be simply obtained by different random weight initializations.

The Squad results (table 2) should list a more recent SOTA result to be fair as it gives the impression that the system presented here is SOTA.

---

### Official Review · AnonReviewer3 · 2017-11-27
**Not well-designed structure and less meaningful discussion**

**Rating:** 4
**Confidence:** 4

**Review:**

This paper proposes a method to find an effective structure of RNNs and attention mechanisms by searching programs over the stack-oriented execution engine.

Although the new point in this paper looks only the representation paradigm of each program: (possibly variable length) list of the function applications, that could be a flexible framework to find a function without any prior structures like Fig.1-left.

However, the design of the execution engine looks not well-designed. E.g., authors described that the engine ignores the binary operations that could not be executed at the time. But in my thought, such operations should not be included in the set of candidate operations, i.e., the set of candidates should be constrained directly by the state of the stack.
Also, including repeating "identity" operations (in the candidates of attention operations) seems that some unnecessary redundancy is introduced into the search space. The same expressiveness could be achieved by predicting a special token only once at the end of the sequence (namely, "end-of-sequence" token as just same as usual auto-regressive RNN-based decoder models).

Comparison in experiments looks meaningless. Score improvement is slight nevertheless authors paid much computation cost for searching accurate network structures. The conventional method (Zoph&Le,17) in row 3 of Table 1 looks not comparable with proposed methods because it is trained by an out-of-domain task (LM) using conventional (tree-based) search space. Authors should at least show the result by applying the conventional search space to the tasks of this paper.
In Table 2, the "our baseline" looks cheap because the dot product is the least attention model in those proposed in past studies.

The catastrophic score drop in the rows 5 and 7 in Table 1 looks interesting, but the paper does not show enough comprehension about this phenomenon, which makes the proposed method hard to apply other tasks.
The same problem exists in the setting of the hyperparameters in the reward functions. According to the footnote, there are largely different settings about the value of \beta, which suggest a sensitivity by changing this parameter. Authors should provide some criterion to choose these hyperparameters.

---

### Official Review · AnonReviewer1 · 2017-12-01
**Computational power**

**Rating:** 3
**Confidence:** 4

**Review:**

This paper experiments the application of NAS to some natural language processing tasks : machine translation and question answering.

My main concern about this paper is its contribution. The difference with the paper of Zoph 2017 is really slight in terms of methodology. Moving from a language modeling task to machine translation is not very impressive neither really discussed. It could be interesting to change the NAS approach by taking into account this application shift.

On the experimental part, the paper is not really convincing. The results on WMT are not state of the art. The best system of this year was a standard phrase based and has achieved 29.3 BLEU score (for BLEU cased, otherwise it's one point more). Therefore the results on mt tasks are difficult to interpret.

At the end , the reader can be sure these experiments required a significant computational power. Beyond that it is difficult to really draw meaningful conclusions.

---

### Public Comment · (anonymous) · 2017-11-13
**Poor evaluation**

Given that the original PTB NAS claims of outperforming LSTMs have been thoroughly debunked by a hyperparameter optimizer with 5x less compute [1], and that hyperparameter optimization is doing qualitatively the same thing as NAS, it'd be good to mention it somewhere in the related work/intro.

Showing marginal improvements off of a very weak SQUAD baseline isn't terribly impressive. According to the latest leaderboard [2], their baseline model, BiDAF from a year ago, is ranked 34th, and their improved model would be 24th (the same as the new BiDAF simple baseline, and 5.4 F1 points below SOTA). Perhaps you should include some more results here to more accurately represent your findings.

Can you give a meaningful comparison of the compute for your NAS versus the amount used for the GNMT baseline? All I see is a mention of using 100-200 GPUs for some unspecified period of time. If you're using more compute than GNMT, can you take that into account for that to produce a meaningful comparison?

[1] https://openreview.net/forum?id=ByJHuTgA-
[2] https://rajpurkar.github.io/SQuAD-explorer/

---

### Author Response · Authors · 2018-01-05
**Thanks the reviewers and the public for the comments**

Dear reviewers and the public,

We would like to thank the reviewers and the public again for weighing in the paper. We will try to improve the evaluation in the next revision of the paper.

Authors

---

### Decision · Program_Chairs · 2018-01-29
**ICLR 2018 Conference Acceptance Decision**

**Decision:**

Reject

**Comment:**

This paper extends work on neural architecture search by introducing a new framework for searching and experiments on new domains of NMT and QA. The results of the work are  beneficial and show improvements using this approach. However the reviewers point out significant issues with the approach itself:

- There is skepticism about the use of NAS in general, particular compared to using the same computational power for other types of simpler hyperparameter search.
- There is general concern about the use of such large scale brute force methods in general. Several of the reviewers expressed concerns about ever possibly being able to replicate these results.
- Given the computational power required, the reviewers feel like the gains are not particularly large, for instance the Squad results not being compared to the best reported systems.